# Improved Dynamic Algorithm for Non-monotone Submodular Maximization under Cardinality Constraint

**Kiarash Banihashem** [* 1]   **Samira Goudarzi** [* 1]   **MohammadTaghi Hajiaghayi** [* 1]   **Peyman Jabbarzade** [* 1]
**Morteza Monemizadeh** [* 2]

## Abstract

We study fully dynamic non-monotone submodular maximization under a cardinality constraint $k$. Prior work achieved approximation guarantees of $(0.125 - \epsilon)$ using $\tilde{O}(\epsilon^{-1}k^2)$ oracle queries per update (NeurIPS'20) and 0.171 using $\tilde{O}(\epsilon^{-3}k^4)$ oracle queries per update (NeurIPS'25). In this work, we present a dynamic algorithm that achieves a 0.262-approximation with worst-case expected update time $O(\epsilon^{-3}k\log(k)\log(\epsilon^{-1}k) + \epsilon^{-2}k^2\log(k))$, where $0 < \epsilon \leq 1$ is the error parameter. We also develop another dynamic algorithm with update time bounded by $\text{poly}(\epsilon^{-1}, k)$ that achieves a 0.277-approximation guarantee.

## 1. Introduction

Submodular functions are essential for addressing a range of real-world challenges, offering a theoretical basis for the "diminishing returns" phenomenon observed in practice (Fujishige, 1984). Many theoretical problems involving graph cuts, coverage, and mutual information can be addressed through submodular maximization. Consequently, submodular functions are increasingly applied in machine learning tasks such as data summarization (Simon et al., 2007; Tschiatschek et al., 2014; Sipos et al., 2012), feature selection (Khanna et al., 2017; Das & Kempe, 2008; 2018), and recommendation systems (El-Arini & Guestrin, 2011).

The general problem of non-monotone submodular maximization has been investigated in various studies (Buchbinder et al., 2015; Balkanski et al., 2018; Buchbinder et al., 2014). This problem is relevant to numerous applications, such as video summarization, movie recommendation (Mirzasoleiman et al., 2016), and revenue maximization in viral marketing (Hartline et al., 2008). A key application of non-monotone submodular maximization involves

optimizing the trade-off between a monotone submodular function and a linear penalty function that discourages the inclusion of additional elements. In particular, in scenarios like maximum facility location, where the goal is to open a subset of facilities to maximize total profit from served clients while accounting for the costs of unopened facilities (Dueck & Frey, 2007). Another application is in machine learning, particularly in feature selection, where learning problems can be framed using weakly submodular functions (Elenberg et al., 2016; Khanna et al., 2017).

In this paper, we address the problem of non-monotone submodular maximization under a cardinality constraint $k$ in the *fully dynamic setting*. In this model, we have a ground set $V$. At any time $t$, the active set $V_t \subseteq V$ consists of elements that have been inserted and not subsequently deleted. We assume a non-monotone submodular function $f$ is defined over $V$. Our objective is to maintain, at any point in time, a subset of $V_t$ of size at most $k$ that maximizes the submodular value.

Finding such a subset is known to be NP-hard (Feige et al., 2011), even in the offline setting where all items are given. Consequently, we focus on developing algorithms with provable approximation guarantees and efficient update times. This is challenging because elements may be inserted or deleted in an adversarial manner. While there are several dynamic algorithms for monotone submodular maximization, the non-monotone variant is more complex since adding elements can decrease the value of a set. In STOC 2022, (Chen & Peng, 2022) posed the open question: "*Can we extend [fully dynamic] results (algorithm or hardness) to non-monotone submodular maximization?*"

Recently, (Banihashem et al., 2023) introduced the first dynamic algorithms for non-monotone submodular maximization. They proposed a dynamic algorithm that maintains a $(0.125 - \epsilon)$-approximate solution using an expected amortized $O(\epsilon^{-1}k^2\log^3 k)$ oracle queries per update. This was subsequently improved by Banihashem et al. (Banihashem et al., 2025) to a $(0.171 - \epsilon)$-approximation, albeit at the cost of $O(\epsilon^{-3}k^4\log^2 k)$ oracle queries per update.

Despite these advances, a substantial gap remains between the approximation factor of the best streaming algorithm for

---

[1]Department of Computer Science, University of Maryland, MD, USA [2]Department of Mathematics and Computer Science, TU Eindhoven, the Netherlands. Correspondence to: Morteza Monemizadeh <m.monemizadeh@tue.nl>.

*Proceedings of the 43$^{rd}$ International Conference on Machine Learning*, Seoul, South Korea. PMLR 306, 2026. Copyright 2026 by the author(s).

non-monotone submodular maximization with a cardinality constraint and that of the dynamic algorithms. In particular, the streaming algorithm of (Alaluf et al., 2022) achieves a 0.277-approximation, significantly higher than the $(0.171 - \epsilon)$-approximation of the current dynamic algorithms. This naturally raises the question of whether a dynamic algorithm can be developed that matches the 0.277-approximation guarantee of (Alaluf et al., 2022).

In this paper, we show that this goal is indeed achievable. Building on the dynamic framework of (Banihashem et al., 2024) for monotone submodular maximization under matroid constraints, we design a dynamic algorithm for the non-monotone setting that effectively simulates the streaming algorithm of (Alaluf et al., 2022) within a fully dynamic environment. More specifically, by using the offline algorithm of (Buchbinder & Feldman, 2019), we obtain a dynamic algorithm for non-monotone submodular maximization under a cardinality constraint $k$ that achieves a 0.277-approximation with worst-case expected update time bounded by $\text{poly}(\epsilon^{-1}, k)$.

To highlight the importance of our result, it is worth noting that while monotone submodular maximization with a cardinality constraint benefits from a $(1 - \frac{1}{e})$-approximation algorithm in the offline setting, and $(0.5 - \epsilon)$-approximation algorithms for both streaming and dynamic scenarios, the non-monotone version faces significant challenges. Specifically, it has been shown by (Gharan & Vondrák, 2011) that achieving a 0.491-approximation is infeasible even in the offline case. In contrast, the best-known algorithms for the non-monotone problem offer approximation factors of 0.385 and 0.277 in offline (Buchbinder & Feldman, 2019) and streaming settings (Alaluf et al., 2020), respectively.

### 1.1. Related Work
The exploration of non-monotone submodular maximization was first addressed by (Feige et al., 2011). They examined the problem of unconstrained non-monotone submodular maximization and developed approximation algorithms for it. They demonstrated that a randomly selected subset $S$ achieves a 0.25-approximation for general functions and a 0.5-approximation for symmetric functions, measured relative to the optimal value in the offline query model. They also introduced two local search strategies: one that achieves a $\frac{1}{3}$-approximation using function $f$ directly, and another that employs a perturbed version of $f$ to attain a better approximation of 0.4 for maximizing unconstrained non-monotone submodular functions. Indeed, they demonstrated that achieving a $(0.5 + \epsilon)$-approximation for symmetric submodular functions necessitates an exponential number of queries for any fixed $\epsilon > 0$.

In subsequent work, (Gharan & Vondrák, 2011) extended the 0.4-approximation algorithm, interpreting it as a simu-

lated annealing approach that improves the approximation to approximately 0.41, and obtained a 0.325-approximation for submodular maximization under a matroid constraint. Later, (Buchbinder et al., 2015) developed a randomized linear-time algorithm for unconstrained maximization that achieves a tight 0.5-approximation, matching the hardness result established in (Feige et al., 2011). Additionally, (Bateni et al., 2013) and (Gupta et al., 2010) independently explored non-monotone submodular maximization with a cardinality constraint $k$ in offline and secretary models, with Gupta et al. achieving an offline approximation factor of 0.153.

While these algorithms are designed for offline scenarios, the need for real-time processing of evolving data streams has led to investigations in dynamic models. Insertion-only streaming algorithms (Mirzasoleiman et al., 2018; Feldman et al., 2018; 2011) do not handle deletions. Although some streaming algorithms for monotone submodular maximization support deletions (Kazemi et al., 2018; Mirzasoleiman et al., 2017), their space and update times scale with the number of deletions, which can be as large as $\Omega(n)$, where $n$ is the size of the ground set $V$.

For monotone submodular maximization, (Badanidiyuru et al., 2014) proposed an insertion-only streaming algorithm with a $(0.5 - \epsilon)$-approximation guarantee under a cardinality constraint $k$. (Chekuri et al., 2015) introduced streaming algorithms for both monotone and non-monotone submodular functions subject to $p$-matchoid constraints. (Mirzasoleiman et al., 2018) and (Feldman et al., 2018) later developed improved streaming algorithms for maximizing non-monotone functions under $p$-matchoid constraints. Currently, the best known streaming algorithm for non-monotone submodular maximization under a cardinality constraint is by (Alaluf et al., 2020), which achieves an approximation factor of 0.277, improving upon the 0.171-approximation previously obtained by (Feldman et al., 2018).

In the dynamic setting, monotone submodular maximization with a cardinality constraint $k$ was first investigated by (Lattanzi et al., 2020) and (Monemizadeh, 2020) at NeurIPS'20. Both approaches developed dynamic algorithms that maintain $(0.5 - \epsilon)$-approximate solutions with efficient query times. Specifically, the algorithm (Lattanzi et al., 2020) has an expected amortized query complexity of $O(\epsilon^{-11} \log^6(k) \log^2(n))$, while the algorithm of (Monemizadeh, 2020) achieves an amortized query complexity of $O(\epsilon^{-3} k^2 \log^5(n))$. (Chen & Peng, 2022) later demonstrated that developing a $c$-approximation dynamic algorithm for $c > 0.5$ would require a polynomial number of oracle queries proportional to the size of the ground set $V$.

More recently, (Duetting et al., 2023) examined monotone submodular maximization under a matroid constraint in the dynamic setting, proposing a $(0.25 - \epsilon)$-approximation al-

gorithm with an amortized expected query complexity of $O(\frac{k^2}{\epsilon} \log(k) \log^2(n) \log^3(\frac{k}{\epsilon}))$. Simultaneously, (Banihashem et al., 2024) achieved similar approximation guarantees with improved query complexities, specifically providing a worst-case expected query complexity of $O(k \log(k) \log^3(\frac{k}{\epsilon}))$. They also enhanced the query complexity of Monemizadeh's dynamic algorithm for cardinality constraints to an expected $O(k\epsilon^{-1} \log^2(k))$.

For non-monotone submodular maximization in dynamic settings, the first dynamic algorithm was given by (Banihashem et al., 2023)(NeurIPS'20), achieving a $(0.125 - \epsilon)$-approximation with expected amortized $\tilde{O}(\epsilon^{-1}k^2)$ oracle queries per update. This was later improved by (Banihashem et al., 2025)(NeurIPS'25) to a 0.171-approximation using $\tilde{O}(\epsilon^{-3}k^4)$ oracle queries per update.

### 1.2. Preliminaries

**Definition 1** (Submodular function). *Let $V$ be a ground set of elements. We define a function $f : 2^V \to \mathbb{R}_{\geq 0}$ as submodular if for any subsets $A, B \subseteq V$ it satisfies $f(A) + f(B) \geq f(A \cup B) + f(A \cap B)$.*

This condition implies that adding an element to a smaller subset provides at least as much marginal gain as adding it to a larger subset. Equivalently, for any subsets $A \subseteq B \subseteq V$ and an element $e \in V \setminus B$, submodularity requires: $f(A \cup \{e\}) - f(A) \geq f(B \cup \{e\}) - f(B)$. We define the *marginal gain* of adding element $e$ to set $A$ as: $f(e \mid A) := f(A \cup \{e\}) - f(A)$. Similarly, for any sets $A, B \subseteq V$, we define: $f(B \mid A) := f(A \cup B) - f(A)$. A function $f$ is called *monotone* if $f(A) \leq f(B)$ for any $A \subseteq B \subseteq V$. In contrast, a function is *non-monotone* if this property does not necessarily hold.

**Definition 2** (Submodular maximization problem with a cardinality constraint $k$). *In this problem, the goal is to find a subset $S^*$ such that $|S^*| \leq k$ and $f(S^*) = \max_{|S| \leq k} f(S)$. Here, $f$ is a submodular function, and $k$ is a positive integer specifying the maximum size of the subset.*

For dynamic analysis, as explored in recent works (Lattanzi et al., 2020; Monemizadeh, 2020; Chen & Peng, 2022; Duetting et al., 2023), we assume access to an oracle for $f$, which allows performing set queries. For any subset $A \subseteq V$, the value $f(A)$ can be retrieved, and the marginal gain $f(e \mid A) \doteq f(A \cup e) - f(A)$ is computed using two set queries: $f(A \cup e)$ and $f(A)$.

Consider a sequence of operations, $\mathcal{S}$, consisting of insertions and deletions applied to the universe $V$. We assume $f : 2^V \to \mathbb{R}_{\geq 0}$ is a (potentially non-monotone) submodular function defined on subsets of $V$. The time $t$ corresponds to the $t^{\text{th}}$ update (either an insertion or deletion) in the sequence. Let $\mathcal{S}_t$ denote the subsequence of updates from the beginning of $\mathcal{S}$ up to time $t$, and let $V_t \subseteq V$ represent the set of elements that have been inserted but not deleted

up to time $t$. Thus, $V_t$ constitutes the current ground set of elements. We denote the optimal value at time $t$ by $OPT_t = \max_{S \subseteq V_t : |S| \leq k} f(S)$.

**Definition 3** (Query complexity). *The query complexity of a dynamic $\alpha$-approximate algorithm refers to the number of oracle queries required by the algorithm to compute a solution $S_t$ with respect to the ground set $V_t$, such that $|S_t| \leq k$ and $f(S_t) \geq \alpha \cdot OPT_t$.*

It is important to note that dynamic algorithms maintain a record of all previous queries, which can potentially be utilized to find $S_t$ at the current time $t$. The dynamic algorithms that we develop operate under the *oblivious adversarial model*, a common framework for analyzing randomized data structures like universal hashing (Carter & Wegman, 1977). In this model, the adversary is aware of the submodular function $f$ and the algorithm being used, allowing it to determine the sequence of insertions and deletions. However, the adversary cannot adapt its strategy based on the algorithm's random choices and thus cannot alter the updates dynamically in response to these choices. This is equivalent to assuming that the adversary prepares the entire sequence of updates before the algorithm begins its execution.

## 2. Dynamic Algorithm

**Setting.** In our algorithm, we assume having access to an offline $\alpha$−approximation algorithm denoted by OFFLINEALG, which can solve the problem of maximizing a non-monotone submodular function $f$ under the cardinality constraint $k$ on any ground set $U$ using $g(|U|, k)$ oracle queries. The algorithm by Buchbinder & Feldman (2019), offering the best known approximation guarantee, or a faster alternative such as that of Buchbinder et al. (2014), could be used as OFFLINEALG. Also, throughout our algorithm, we use a fixed constant in $(0, 1]$ as $\epsilon$, and set $p$ to $\lceil \frac{4}{\epsilon} \rceil$.

**Parallel Runs.** To solve non-monotone maximization in the dynamic setting, we use the technique of maintaining parallel runs. For each run $i$, we designate a guess for the optimal solution as $\tau = (1 + \epsilon)^i$. We later prove that at any time $t$, the approximation guarantee holds for the solution of the run with a $\tau$ close to the optimal solution at time $t$ and obeys $(\frac{1}{1+\epsilon})OPT_t \leq \tau \leq OPT_t$. Hence, by choosing the best solution among the maintained solutions, we would achieve the guaranteed approximation ratio at all times.

As stated above, in our proof for the approximation guarantee, we are concerned with a run only when its $\tau$ parameter is at least $(\frac{1}{1+\epsilon})OPT_t$, which wouldn't happen unless all of the present elements at time $t$ have $f(e) \leq \tau(1 + \epsilon)$. Therefore, a run with parameter $\tau = (1 + \epsilon)^i$ can ignore any element with $f(e)$ greater than $(1 + \epsilon)^{i+1}$ without affecting the approximation guarantee. Hence, any update (insertion or deletion)

related to an element $e$ needs to be handled by instance $i$ only if $i + 1 \geq \log_{1+\epsilon}(f(e))$ or equivalently $i \geq \log_{1+\epsilon}(\frac{f(e)}{1+\epsilon})$.

On the other hand, a run with parameter $\tau$ uses the threshold $\frac{c\tau}{k}$ for checking the suitability of elements and the filtering process, and the submodularity of the function $f$ guarantees that an element $e$ with $f(e) < \frac{c\tau}{k}$ would be useless for that run and can be discarded. Therefore, for any run with $\tau = (1+\epsilon)^i$ we only need the elements with $f(e) \geq \frac{c(1+\epsilon)^i}{k}$, which means an update related to an element $e$ only needs to be handled by the $i^{\text{th}}$ run if $\log_{1+\epsilon}(\frac{k \cdot f(e)}{c}) \geq i$.

Algorithm 1 sets the value of the parameter $c$, instantiates the parallel runs, and after any insertion or deletion changes the set of currently present elements and passes on the update to appropriate instances based on the ranges specified above.

---

**Algorithm 1** PARALLELRUNS $(p, \alpha)$

---

1: **function** INITIALIZE
2:     $c \leftarrow \frac{\alpha}{1+\alpha}$
3:    **for** $i \in \mathbb{Z}$ **do**
4:       Let $\mathcal{I}_i$ be the instance of our dynamic algorithm, for which $\tau = (1+\epsilon)^i$.
5: **function** GLOBALINSERT$(e)$
6:     $V \leftarrow V \cup \{e\}$
7:    **for** $\log_{1+\epsilon}(\frac{f(e)}{1+\epsilon}) \leq i \leq \log_{1+\epsilon}(\frac{k \cdot f(e)}{c})$ **do**
8:       Invoke INSERT$(e)$ for instance $\mathcal{I}_i$.
9: **function** GLOBALDELETE$(e)$
10:    $V \leftarrow V \backslash \{e\}$
11:    **for** $\log_{1+\epsilon}(\frac{f(e)}{1+\epsilon}) \leq i \leq \log_{1+\epsilon}(\frac{k \cdot f(e)}{c})$ **do**
12:       Invoke DELETE$(e)$ for instance $\mathcal{I}_i$.

---

**Structure of Levels.** Each instance of our dynamic algorithm with a fixed parameter $\tau$ maintains a randomized leveled data structure that would concurrently build $p$ distinct potential solutions.

Consider a data structure consisting of $T$ main levels, and each level $1 \leq \ell \leq T$ consists of $R_\ell$, $e_\ell$, $x_\ell$, and $p$ solution sets $S_{1,\ell}, S_{2,\ell}, \ldots, S_{p,\ell}$, described below.

For each level $1 \leq \ell \leq T$, $R_\ell$ includes all the elements that are suitable for at least one of the $p$ solutions of the previous level, where being suitable for a solution means the potential of increasing its value at least by the threshold $\frac{c\tau}{k}$ while not exceeding the cardinality constraint $k$. It should be noted that by solutions of the previous level for $\ell = 1$, we mean $p$ empty sets. Among the $p$ solution sets $S_{1,\ell}, S_{2,\ell}, \ldots, S_{p,\ell}$, $p - 1$ of them are exactly as they were in the previous level, and one of them has exactly one more element in it. $e_\ell$ is the element added to the solutions at level $\ell$, and $x_\ell$ is the index of the solution $e_\ell$ is added to, which means $e_\ell$ is the extra element in $S_{x_\ell,\ell}$ compared to $S_{x_\ell,\ell-1}$. Additionally, for level $\ell = T$, There is no suitable element for any of its $p$ solutions.

As stated above, for each level $\ell$, the content of $R_\ell$ depends on the solutions of its previous levels, which is also why the choice of $e_\ell$ affects all subsequent levels.

Now, consider having constructed the whole data structure as described before. In order to keep the data structure as described, if we are to change our choice of $e_\ell$ for level $\ell$, we would also need to reconstruct all of its subsequent levels. Therefore, if the adversary knows an element is likely to be in a solution, they can make us reconstruct part of our data structure. This is the reason behind the randomness aspect of our data structure.

We make sure that for any level $\ell$, $e_\ell$ would be a uniformly random element of $R_\ell$, given its previous levels. This means that conditioned on the first $\ell - 1$ levels, which also deterministically determines $R_\ell$, $e_\ell$ would be any element of $R_\ell$ with probability of $\frac{1}{|R_\ell|}$. This quality, which we call *uniform invariant*, will bound the likeliness of needing to reconstruct part of our data structure after the deletions. However, maintaining it requires random partial reconstructions after insertions.

After having the whole data structure constructed, we'll have $p$ solutions to any of which no other element with a considerable marginal gain can be added. Similar to the algorithm of (Alaluf et al., 2022), we run the OFFLINEALGON the combination of these solutions, to obtain $S_{\text{off}}$, and the best set among $S_{1,T}, S_{2,T}, \ldots, S_{p,T}$ and $S_{\text{off}}$ would be the solution of this instance of our algorithm, which is guaranteed to achieve our approximation ratio if its $\tau$ is a correct estimation of OPT.

**Construction of Levels.** We assume the first $i - 1$ (possibly 0) levels of an instance are already constructed and $R_i$ is the set of all the elements that are suitable for at least one of the $p$ solutions of level $i - 1$ ($p$ empty sets if $i = 1$), and we want to build upon them to complete the data structure and finally find the solution of the instance based on $S_{1,T}, S_{2,T}, \ldots, S_{p,T}$.

Note that the Algorithm 1 passes along updates related to an element to an instance only if that element is suitable for an empty set in that instance, so the assumption of having a correct $R_1$ is always valid.

To construct the rest of the data structure, we begin by choosing a random permutation of $R_i$ that we denote by $P$. In this way for each $A \subseteq R_i$, any of the elements in $A$ are equally likely to be the first element of $A$ that appears in the permutation $P$. We then set $\ell$ to be $i$, which will always have the index of the last level that we have started to build. We iterate over the elements of $P$, and for each element $e$ we check if it is suitable for any of the solutions of $S_{1,\ell-1}, S_{2,\ell-1}, \ldots, S_{p,\ell-1}$ by setting $x$ to be the output of CHECKSUITABLE$(\ell - 1, e)$. We next consider two cases.

- If $e$ is suitable for a non-empty subset of them, $x$ would be the index of the lowest indexed solution among them which is a number between 1 and $p$. We set $S_{j,\ell}$ equal to $S_{j,\ell-1}$ by default for every $1 \le j \le p$, we add $e$ to $S_{x,\ell}$, and we set $e_\ell$ and $x_\ell$ to be $e$ and $x$, respectively to remember the selected element of level $\ell$ and the index of the solution to which it was added. After doing so, we have assigned the value of every variable related to level $\ell$ other than $R_\ell$ (if $\ell > i$). If $\ell > i$, we add each of the elements that should be in $R_\ell$ by definition to $R_\ell$ right before we finish processing that element. Specifically, because of the submodularity of the function $f$, we know that the element $e$ should be added to $R$ of all levels between $i + 1$ and $\ell$ and none of the subsequent levels. Thus, we let $z$ be $\ell$ so that it gets added to $R_{i+1}, \ldots, R_\ell$ in line 26.

  Now that we are all set with level $\ell$, we begin building the next level by increasing $\ell$ by one and defining the new $R_\ell$, which is initialized to be an empty set.

- If $e$ is not suitable for any of the solutions of $S_{1,\ell-1}, S_{2,\ell-1}, \ldots, S_{p,\ell-1}$ and $x = 0$, we add it to $R$ for all appropriate levels based on the definition. By the submodularity of function $f$, there exists a level $z$ where $e$ is suitable for at least one of the $p$ solutions of any level before $z$ and not suitable for any of $p$ solutions of level $z$ or any level after it. We can find $z$ using binary search. Since $z$ is the last level where $e \in R_z$, so we add $e$ to $R_{i+1}, \ldots, R_z$ in line 26.

When we are finished processing all the elements, all of the levels between $i$ and $\ell - 1$ are completely built, and the last level that we started to build would have remained empty. Thus, we set $T$ to be $\ell - 1$.

Additionally, based on the explanations on how $e_j$ gets selected for each level $\ell$, we know that conditioned on $e_i, \ldots, e_{j-1}$, $R_j$ can be deterministically determined, and $e_j$ would be the first element of $R_j$ appeared in P, which makes it a random variable uniformly chosen from $R_j$.

Lastly, we run OFFLINEALG on $\cup_{i=0}^{p} S_{i,T}$, and we select the best set among $S_{1,T}, \ldots, S_{p,T}$ and $S_{\text{off}}$ as our solution. Note that since each of $S_{i,T}$ has at most $k$ elements, $\cup_{i=0}^{p} S_{i,T}$ has at most $pk$ elements and $T$ is also at most $pk$ since at each level exactly one element gets added to the $p$ solution of its previous levels.

**Updates.** Now, we explain how an instance handles the insertion or deletion of an element $e$.

**Insertions.** When an element $v$ gets inserted, we iterate through all levels (including the incomplete level $T + 1$). At level $i$, we set $x$ to be the output of CHECKSUITABLE$(i, v)$. If $x = 0$, we no longer need to continue this process and

---

**Algorithm 2** LEVELCONSTRUCTION $(p, \alpha, \tau)$

1: **function** INITIALIZE
2:      $S_{1,0}, S_{2,0}, \ldots, S_{p,0} \leftarrow \emptyset$
3:      $R_1 \leftarrow \emptyset$
4:      Sol $\leftarrow \emptyset$
5: **function** CHECKSUITABLE$(i, e)$
6:      **for** $j \leftarrow 1$ **to** $p$ **do**
7:          **if** $|S_{j,i}| < k$ and $f(e|S_{j,i}) \ge \frac{c\tau}{k}$ **then**
8:              **return** $j$
9:      **return** 0
10: **function** FILTER$(i)$
11:      **return** $\{e \in R_i : \text{CHECKSUITABLE}(i, e) \ne 0\}$
12: **function** CONSTRUCTLEVEL $(i)$
13:      Let $P$ be a random permutation of elements of $R_i$
14:      $\ell \leftarrow i$
15:      **for** $e$ in $P$ **do**
16:          $x \leftarrow \text{CHECKSUITABLE}(\ell - 1, e)$
17:          **if** $x \ne 0$ **then**
18:              **for** $j \leftarrow 1$ **to** $p$ **do**
19:                  $S_{j,\ell} \leftarrow S_{j,\ell-1}$
20:              $S_{x,\ell} \leftarrow S_{x,\ell} \cup \{e\}$ , $e_\ell \leftarrow e$ , $x_\ell \leftarrow x$
21:              $z \leftarrow \ell$
22:              $\ell \leftarrow \ell + 1$ , $R_\ell \leftarrow \emptyset$
23:          **else**
24:              Find the lowest $z \in [i, \ell - 1]$ that CHECKSUITABLE$(z, e) = 0$ by Binary Search
25:              **for** $j \leftarrow i + 1$ **to** $z$ **do**
26:                  $R_j \leftarrow R_j \cup \{e\}$
27:      $T \leftarrow \ell - 1$
28:      $S_{\text{off}} \leftarrow$ The output of OFFLINEALG with $\cup_{i=0}^{p} S_{i,T}$ as its input
29:      Sol $\leftarrow$ The set maximizing $f$ among $S_{1,T}, \ldots, S_{p,T}$ and $S_{\text{off}}$

---

terminate the loop. When $x \ne 0$, we add $v$ to $R_i$. Then to preserve the uniform invariant, with a probability of $\frac{|R_i|-1}{|R_i|}$ we proceed to the next level, and with a probability of $\frac{1}{|R_i|}$ we change the selected element of level $i$ to $v$.

We first change set $S_{j,i}$ to $S_{j,i-1}$ by default for any $1 \le j \le p$. We then change $S_{x_i,i}$, $e_i$, and $x_i$ to $S_{x,i} \cup \{v\}$, $v$ and $x$, respectively. Afterward, when we are done with the changes in the level $i$, we set $R_{i+1}$ to the output of FILTER$(i, v)$. Once we have a correct $R_{i+1}$, we invoke CONSTRUCTLEVEL $(i + 1)$ to reconstruct all the subsequent levels. In this way, conditioned on previous levels, with a probability of $\frac{1}{|R_i|}$, $e_i$ would be $v$, and with a probability of $\frac{|R_i|-1}{|R_i|} \cdot \frac{1}{|R_i|-1} = \frac{1}{|R_i|}$, $e_i$ would be any element in $R_i$ other than $v$. Note that if the for loop reaches $i = T + 1$ and $v$ gets added to $R_{T+1}$, which was previously empty, level $T + 1$ gets built, building of level $T + 2$ start, CONSTRUCTLEVEL $(T + 2)$ gets invoked, and $T$ gets increased.

**Deletions.** When an element $v$ gets deleted, we iterate through all non-empty levels. At level $i$, if $v \notin R_i$, we would know it is not any of the subsequent levels either. Hence, we terminate the loop. Otherwise, we remove $v$ from $R_i$, and we check to see if we had selected $v$ as $e_i$ or not. If $e_i \neq v$, we proceed with the next level, but in the other case, we invoke CONSTRUCTLEVEL $(i)$ to reconstruct the data structure starting from level $i$. Note that $R_i$ is already correct with respect to the previous levels.

This way of handling deletion preserves the uniform invariant. Because for each $i$ one of the following holds:

**(1)** Level $i$ is either constructed as a result of the invocation of CONSTRUCTLEVEL $(j)$ for a $j \leq i$, in which case, conditioned on levels preceding level $i$, $e_i$ is a uniformly random element in $R_i$.

**(2)** Whether $R_i$ used to include $v$ or not, $R_i$ has become the past version of $R_i$ without $v$. In other words, if we denote the past version of $R_i$ with $R_i^-$, we know that $R_i = R_i^- \setminus \{v\}$. We also know that $e_i$ was not $v$. We knew beforehand that for any two elements $a$ and $b$ in $R_i^-$, the probability of $e_i$ being $a$ and $e_i$ being $b$ are equal. We know that the probability of $e_i$ being $v$ is 0. Yet for any two elements $a$ and $b$ in $R_i^- \setminus \{v\} = R_i$, the probability of $e_i$ being either $a$ or $b$ should have remained equal. This means that conditioned on levels before $i$, $e_i$ is a uniformly chosen random element from $R_i$.

Note that, in the case that the only element in last level is $v$, $R_T$ becomes an empty set, CONSTRUCTLEVEL $(T)$ gets invoked and $T$ becomes $T - 1$.

For both insertion and deletion, it should be noted that if all of the $p$ solutions remain unchanged, both $S_{\text{off}}$ and Sol remain valid, and any change in the solutions will result in a partial reconstruction, which ends with an automatic recalculation of $S_{\text{off}}$ and Sol.

## 3. Analysis

In this section, we prove the following Theorem, which follows from a combination of Theorems 9 and 14.

**Theorem 4.** *At each time, the output of our algorithm satisfies an approximation guarantee of $\frac{\alpha}{1+\alpha} - \epsilon$, and the expected query complexity of each update is at most $O\big(\log(k)\,\epsilon^{-1}(\epsilon^{-3}k\log(k\epsilon^{-1}) + g(k\epsilon^{-1}, k))\big)$, where $g(|U|, k)$ denotes the number of oracle calls required by the OFFLINEALG to maximize a non-monotone submodular function $f$ under a cardinality constraint $k$ on a ground set $U$.*

By utilizing the SODA'14 algorithm of Buchbinder et al. (2014) and setting $\alpha = 0.356$, Theorem 9 yields an approximation guarantee of $(0.26253 - \epsilon)$ with an expected query complexity of $O(\epsilon^{-3}\log(k)\log(\epsilon^{-1}k) + \epsilon^{-2}k^2\log(k))$.

Furthermore, by using the state-of-the-art (in terms of approximation guarantee) algorithm of Buchbinder & Feld-

---

**Algorithm 3** UPDATE$(p, \alpha, \tau)$

1: **function** DELETE$(v)$
2:   **for** $i \leftarrow 1$ **to** $T$ **do**
3:     **if** $v \notin R_i$ **then**
4:       **break**
5:     $R_i \leftarrow R_i \setminus \{v\}$
6:     **if** $e_i = v$ **then**
7:       **return** CONSTRUCTLEVEL $(i)$
8: **function** INSERT$(v)$
9:   **for** $i \leftarrow 1$ **to** $T + 1$ **do**
10:    $x \leftarrow$ CHECKSUITABLE$(i, v)$
11:    **if** $x = 0$ **then**
12:      **break**
13:    $R_i \leftarrow R_i \cup \{v\}$.
14:    Let $p = 1$ with probability $\frac{1}{|R_i|}$, and otherwise $p = 0$
15:    **if** $p = 1$ **then**
16:      **for** $j \leftarrow 1$ **to** $p$ **do**
17:        $S_{j,\ell} \leftarrow S_{j,\ell-1}$
18:      $S_{x,i} \leftarrow S_{x,i-1} \cup \{v\}$, $e_i \leftarrow v$, $x_i \leftarrow x$
19:      $R_{i+1} \leftarrow$ FILTER$(i, v)$
20:      **return** CONSTRUCTLEVEL $(i + 1)$

---

man (2019) and setting $\alpha = 0.385$, Theorem 9 yields an approximation of $(0.27797 - \epsilon)$ with an expected poly$(\epsilon^{-1}, k)$ query complexity per update. This is because $g(|U|, k) = O(\text{poly}(|U|, k))$ ensures an $O(\text{poly}(\epsilon^{-1}, k))$ update time for our algorithm.

**Approximation Guarantee.** We refer to Algorithm 1 presented in (Alaluf et al., 2022) as STREAMPROCESS. This algorithm also assumes having access to an offline algorithm denoted by OFFLINEALG, which solves the non-monotone submodular maximization problem under the cardinality constraint $k$ with an approximation ratio of $\alpha$.

The STREAMPROCESS algorithm sets the parameter $c$ to be $\frac{\alpha}{\alpha+1}$, and maintains $p$ solutions $S_1, \ldots, S_p$, which are initially empty sets. It processes the elements of the stream one by one, and for each arriving element $e$, checks if there is any solution $S_i$, for which $e$ is a suitable element, meaning that it has less than $k$ elements and $f(e|S_i) \geq \frac{c\tau}{k}$. If there is one or more such solutions, it adds $e$ an arbitrary one of them. Afterward, when every element is processed, STREAMPROCESS sets $S_{\text{off}}$ to be the output of the OFFLINEALGalgorithm on the union of $S_1, \ldots, S_p$, and chooses the best solution among $S_{\text{off}}$ and $S_1, \ldots, S_p$ as its output.

**Lemma 5** (Proposition 8 of (Alaluf et al., 2022)). *Given any fixed $p$ and $\tau$, the output set of STREAMPROCESS has an expected value of at least $(\frac{\alpha}{1+\alpha} - \frac{2}{p}) \cdot \tau$.*

*Proof.* For a complete proof, please check pages 7 to 10 of (Alaluf et al., 2022). □

Now consider a slightly modified version of STREAMPROCESS, that for each element $e$ instead of adding $e$ to a solution chosen arbitrarily among all the solutions that $e$ is suitable for them, adds $e$ to the solution with the lowest index among all those solutions. Given that the proofs of STREAMPROCESS in (Alaluf et al., 2022) work for any arbitrary selection, they would work for this modified version. From this point on, whenever we refer to STREAMPROCESS, we mean this modified version of STREAMPROCESS.

Now, we will demonstrate that after any GLOBALINSERTand GLOBALDELETE, for any instance $\mathcal{I}_i$, there exists a stream $\mathcal{S}$ of elements in $R_1$ of $\mathcal{I}_i$ such that Sol of $\mathcal{I}_i$ matches the output of STREAMPROCESS on $\mathcal{S}$ with parameters $p$ and $\tau$, proving that our algorithm achieves the same approximation factor as (Alaluf et al., 2022).

**Lemma 6.** *Consider a fixed instance of our algorithm after a fixed update. Let $\mathcal{S}$ be the stream of $e_1, \ldots, e_T$ followed by other elements in $R_1$ in any order. Then $(S_1, \ldots, S_p)$ after running the STREAMPROCESS algorithm on $\mathcal{S}$ with parameters $p$ and $\tau$ would be same as $(S_{1,T}, \ldots, S_{p,T})$.*

*Proof.* We first divide the stream $\mathcal{S}$ into two sub-streams $\mathcal{S}_1$ and $\mathcal{S}_2$, where $\mathcal{S}_1 = e_1, e_2, \ldots, e_T$ in the specified order, and $\mathcal{S}_2$ is the stream containing the elements in $R_1$ that are not in $\mathcal{S}_1$ in an arbitrary order. For each $1 \leq i \leq T$, just before the STREAMPROCESS algorithm processes $e_i$, the state $(S_1, \ldots, S_p)$ matches $(S_{1,i-1}, \ldots, S_{p,i-1})$. The algorithm then adds $e_i$ to $S_{x_i}$, updating the state to $(S_{1,i}, \ldots, S_{p,i})$. Consequently, once $\mathcal{S}_1$ has been fully processed, the state becomes $(S_{1,T}, \ldots, S_{p,T})$. Next, $\mathcal{S}_2$ is processed, which we show does not affect any of the solutions, thus completing our proof.

Consider any element $e \in \mathcal{S}_2$. We know that $e$ had been filtered out at some level since $e \in R_1$ but $e \notin R_T$. Assume that $e$ has been filtered from level $i$, i.e., $e \in R_i$ and $e \notin R_{i+1}$. For each $j \in [1, p]$, we have $S_{j,i} \subseteq S_{j,T}$ implying that $|S_{j,i}| \leq |S_{j,T}|$ and $f(e|S_{j,i}) \geq f(e|S_{j,T})$ because of the submodularity of the function $f$. Hence, knowing that $e$ could not be added to $S_{j,i}$ implies that it can not be added to $S_{j,T}$ as well, which means STREAMPROCESS can not add $e$ to $S_j$. Therefore, STREAMPROCESS would not add $e$ to any of its $p$ solutions. □

**Lemma 7.** *Consider a fixed instance of our algorithm after a fixed update. If there exists a stream $\mathcal{S}$ of the elements in our $R_1$ such that $(S_1, \ldots, S_p)$ after running the STREAMPROCESS algorithm on the stream $\mathcal{S}$ is the same as our $(S_{1,T}, \ldots, S_{p,T})$, and if our algorithm and STREAMPROCESS use the same random bits in case the OFFLINEALGis randomized, our Sol would be the same as the output of STREAMPROCESS.*

The combination of the Lemmas 5, 6, and 7 immediately proves the following Corollary.

**Corollary 8.** *Consider a fixed instance of our algorithm after a fixed update. Sol has an expected value of at least $(\frac{\alpha}{1+\alpha} - \frac{2}{p}) \cdot \tau$.*

**Theorem 9.** *Consider a fixed $t$, and the instance with $\tau$ that obeys $(\frac{1}{1+\epsilon})OPT_t \leq \tau \leq OPT_t$, where $OPT_t$ is the optimal solution after the first $t$ updates. Sol of this instance has an expected value of at least $(\frac{\alpha}{1+\alpha} - \epsilon)OPT_t$.*

*Proof.* By Corollary 8, Sol has an expected value of at least $(\frac{\alpha}{1+\alpha} - \frac{2}{p})\tau$. Since $p = \lceil \frac{4}{\epsilon} \rceil \geq \frac{4}{\epsilon}$, we have

$$\mathbb{E}[\text{Sol}] \geq \left(\frac{\alpha}{1+\alpha} - \frac{\epsilon}{2}\right)\tau \geq \left(\frac{\alpha}{1+\alpha} - \frac{\epsilon}{2}\right)\left(\frac{1}{1+\epsilon}\right)OPT_t,$$

where the second inequality follows from the assumption that $(\frac{1}{1+\epsilon})OPT_t \leq \tau$. Since for any $\alpha \in [0, 1]$, and $\epsilon > 0$, we have $(\frac{\alpha}{1+\alpha} - \frac{\epsilon}{2})(\frac{1}{1+\epsilon}) \geq (\frac{\alpha}{1+\alpha} - \epsilon)$, we can conclude that $\mathbb{E}[\text{Sol}] \geq (\frac{\alpha}{1+\alpha} - \epsilon)OPT_t$. □

**Query Complexity.** Finally, we analyze the query complexity of our algorithm. Recall that, as defined earlier, $g(|U|, k)$ denotes the number of oracle calls made by OF-FLINEALG to maximize a non-monotone submodular function $f$ under a cardinality constraint $k$ on a ground set $U$.

**Lemma 10.** *The number of levels $T$ is at most $pk$.*

*Proof.* We know that $\cup_{i=0}^{p} S_{i,0} = \emptyset$. Thus, for $j = 0$, we know that $\left|\cup_{i=0}^{p} S_{i,j}\right| = 0$. For each $1 \leq j \leq T$, $\left|\cup_{i=0}^{p} S_{i,j}\right| = \left|\cup_{i=0}^{p} S_{i,j-1}\right| + 1$ because exactly one of the $p$ solution of level $j$ has one extra element compared to the level $j-1$, and the rest are exactly the same. Therefore, for any $j \leq T$, $\left|\cup_{i=0}^{p} S_{i,j}\right| = j$, which immediately implies that $\left|\cup_{i=0}^{p} S_{i,T}\right| = T$. On the other hand, we know that $\left|\cup_{i=0}^{p} S_{i,T}\right| \leq pk$ since for any $1 \leq i \leq p$, $S_{i,T}$ has at most $k$ elements. Hence, we have $T \leq pk$. □

**Lemma 11.** *Calling CONSTRUCTLEVEL$(i)$ uses $O(|R_i| p \log(pk))$ query calls to complete the data structure, and uses $O(|R_i| p \log(pk) + g(pk, k) + p)$ query calls in total.*

*Proof.* We know that for any $i$ and any element $e$, invoking CHECKSUITABLE$(i, e)$ uses at most $O(p)$ oracle queries. The algorithm CONSTRUCTLEVEL$(i)$ iterates over all elements in $R_i$, invoking CHECKSUITABLE$(\ell, e)$ once per element, causing $O(p)$ query calls. If CHECKSUITABLE$(\ell, e) \neq 0$, no further queries are made; otherwise, we reach Line 24 and invoke CHECKSUITABLEfor $e$ at most $O(\log(\ell - 1)) = O(\log T) = O(\log(pk))$ times by Lemma 10. Thus, for each element $e$, we make at most $O(p \log(pk))$ query calls, and summing over all elements in $R_i$, the total number of queries to complete the data structure is $O(|R_i| p \log(pk))$.

Subsequently, we invoke OFFLINEALG on $\cup_{i=0}^{p} S_{i,T}$, which makes $g(\left|\cup_{i=0}^{p} S_{i,T}\right|, k)$ query calls. We know that

$\left| \cup_{i=0}^{p} S_{i,T} \right| \leq pk$, and $g$ is an increasing function. Hence, the execution of the OFFLINEALG makes at most $g(pk, k)$ query calls. Therefore, the total number of query calls during the execution of CONSTRUCTLEVEL (i) would also be as stated in the statement of the Lemma. □

**Lemma 12.** *For any fixed instance of our algorithm, the expected query complexity of each update operation in Algorithm 3 is at most $O(p^2 k \log(pk) + g(pk, k))$.*

*Proof.* To prove this lemma, we first need a few notations. Indeed, similar to (Banihashem et al., 2024), for random variables and their values, we use bold and non-bold letters. For example, we denote a random variable by $\mathbf{X}$ and its value by $X$. Next, we use random variables $\mathbf{e}_i$, $\mathbf{R}_i$, $\mathbf{T}$, and $\mathbf{H}_i$ to define the *history* of levels. Indeed, we define $H_i = (e_1, \ldots, e_{i-1}, R_i)$ as *history* of levels before $i$, and $\mathbf{H}_i := (\mathbf{e}_1, \ldots, \mathbf{e}_{i-1}, \mathbf{R}_i)$ would be the random variable that corresponds to $H_i$.

To distinguish between each of these random variables and their corresponding values before and after a fixed update, we use the notations $\mathbf{Y}^-$ and $Y^-$ to denote a random variable and its value before insertion or deletion of an element, and keep using $\mathbf{Y}$ and $Y$ to denote them at the current time after the execution of update is complete. As an example, $\mathbf{H}_i^- := (\mathbf{e}_1^-, \ldots, \mathbf{e}_{i-1}^-, \mathbf{R}_i^-)$ is the random variable that corresponds to the history $H_i^- = (e_1^-, \ldots, e_{i-1}^-, R_i^-)$.

Now, we prove the lemma. Indeed, after the insertion of an element $e$, the probability of having invoked CONSTRUCTLEVEL (i) for any $2 \leq i \leq T$, is equal to $\Pr[\mathbf{e}_{i-1} = e | \mathbf{T} \geq i$ and $\mathbf{H}_{i-1} = H_{i-1}] = \frac{1}{|R_{i-1}|} \cdot \mathbb{1}\{e \in R_{i-1}\}$ by Lemma 13. In case of having invoked CONSTRUCTLEVEL (i), by Lemma 11 we know that we have made at most $O(|R_{i-1}| p) + O(|R_i| p \log(pk)) \leq O(|R_{i-1}| p \log(pk))$ query calls to complete the data structure. Therefore, for any $2 \leq i \leq T$, the expected number of queries used for completing the data structure caused by the invocation of CONSTRUCTLEVEL (i), is $\frac{1}{|R_{i-1}|} O(|R_{i-1}| p \log(pk)) = O(p \log(pk))$.

Similarly, after the deletion of an element $e$, the probability of having invoked CONSTRUCTLEVEL (i), for any $1 \leq i \leq T^-$, is equal to $\Pr\left[\mathbf{e}_i^- = e | \mathbf{T}^- \geq i$ and $\mathbf{H}_i^- = H_i^-\right] = \frac{1}{|R_i^-|} \cdot \mathbb{1}\left\{e \in R_i^-\right\} \leq \frac{1}{|R_i^-|} \leq \frac{1}{|R_i|}$, where the first equality follows from Lemma 13 and the last inequality holds because $|R_i^-| \geq |R_i|$. By Lemma 11, for any $1 \leq i \leq T$, CONSTRUCTLEVEL(i) makes at most $O(|R_i| p \log(pk))$ query calls to complete the data structure. Therefore, for any $i \in [1, T]$, the expected number of queries caused by invoking CONSTRUCTLEVEL (i) to complete the data structure is bounded by $\frac{1}{|R_i|} \cdot O(|R_i| p \log(pk)) = O(p \log(pk))$.

After the execution of any update, we know that for any $i \notin [1, T]$, CONSTRUCTLEVEL (i) has not made any query

call to complete the data structure. Thus, by summing over the expected number of queries in case of calling CONSTRUCTLEVEL (i) for $i = [1, T]$, we calculate the expected number of query calls made to complete the data structure.

The the number of levels $T$ is bounded by $pk$ by Lemma 10. Therefore, After each update, we know that the expected number of query calls made to partially reconstruct the data structure is at most $\sum_{i=1}^{T} O(p \log(pk)) \leq O(p^2 k \log(pk))$. After the execution of any update, we know that we have executed OFFLINEALG and selected Sol at most once. Thus, the total number of query calls made in this run is at most $O(p^2 k \log(pk) + g(pk, k) + p) = O(p^2 k \log(pk) + g(pk, k))$. □

**Uniform Invariant.** We now state the lemma ensuring that the uniform invariant is maintained at all times.

**Lemma 13** (Uniform invariant). *For all $i \geq 1$, conditioned on the random variables $\mathbf{H}_i$, the element $e_i$ is selected uniformly at random from the set $R_i$. Formally, $\Pr[\mathbf{e}_i = e | \mathbf{T} \geq i$ and $\mathbf{H}_i = H_i] = \frac{1}{|R_i|} \cdot \mathbf{1}_{\{e \in R_i\}}$.*

This result follows from a combination of Lemmas 49, 56, and 64 in (Banihashem et al., 2024), which collectively establish that the uniform invariant is preserved after each update. We omit the proof here, as it is provided in (Banihashem et al., 2024).

**Theorem 14.** *The expected query complexity of each update for all runs is $O(\log(k)\epsilon^{-1}(\epsilon^{-2} k \log(k\epsilon^{-1}) + g(k\epsilon^{-1}, k)))$.*

*Proof.* Update of each element gets passed down to at most $\log_{1+\epsilon}\left(\frac{k \cdot f(e)}{c}\right) - \log_{1+\epsilon}\left(\frac{f(e)}{1+\epsilon}\right) + 1 = \log_{1+\epsilon}\left(\frac{k(1+\epsilon)}{c}\right) + 1 = \log_{1+\epsilon}\left(\frac{k}{c}\right) + 2 = O\left(\frac{\log\left(\frac{2k}{\alpha}\right)}{\epsilon}\right) = O\left(\frac{\log(k)}{\epsilon}\right)$, instances of our algorithm. Therefore, Lemma 12 proves that the total query complexity is at most $O(\log(k)\epsilon^{-1}(p^2 k \log(pk) + g(pk, k)))$. Since $p = O(\epsilon^{-1})$, the total query complexity of the algorithm per each update would be as claimed. □

## Impact Statement

This paper presents work whose goal is to advance the field of Machine Learning. The nature of our work is mainly theoretical and focus on submodular maximization, and thus its insights might find use in various areas.

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
