# OpenReview forum: "Improved Dynamic Algorithm for Non-monotone Submodular Maximization under Cardinality Constraint"
_ICML.cc/2026/Conference — ICML 2026 regular_

### Official Review · Reviewer_hNix · 2026-03-09

**Soundness:** 2
**Presentation:** 2
**Significance:** 3
**Originality:** 3
**Overall Recommendation:** 4
**Confidence:** 4

**Summary:**

The paper studies non-monotone submodular maximization in a fully dynamic setting: given a ground set V, at every time t there is a set $V_t \subseteq V$ of active elements, that can added or deleted each round, and the goal is to keep for each time t a subset of $V_t$ of cardinality at most $k$ to maximize the value of this subset by some non-monotone submodular function (a function from $2^V$ that satisfies a diminishing return property).

The previous results before the current paper achieved an approximation guarantee of $0.125 - \epsilon$ using $\epsilon^{-1} k^2$ oracle queries up to logarithmic factors and $0.171 - \epsilon$ using $\epsilon^{-3} k^4$ queries. The paper achieves the following results: a dynamic $0.262$ approximation with worst case expected update time of $\epsilon^{-3} \log k \log \epsilon^{-1} k + \epsilon^{-2} k^2 \log k$, and an additional dynamic algorithm with polynomial update time with approximation guarantee somewhat better of $0.277$.

The techniques are a nice blend of several techniques such as parallel threshold guessing, randomized level maintenance, and streaming-to-dynamic simulations, and more.

**Compliance With Llm Reviewing Policy:**

Affirmed.

**Key Questions For Authors:**

What is the true technical novelty beyond combining known ingredients from prior work?

Are the inconsistencies only typos, or is there an actual bug you could have missed?

Could you add details on the proofs missing?

I would consider increasing the score if there were formal proofs for all claims, so correctness could be verified line by line.

**Limitations:**

yes

**Strengths And Weaknesses:**

- Strengths:
- improved results for a semi-fundamental problem, in my opinion.
- impressive control of prior work and applying previous techniques.


weaknesses:
- Technical novelty and contribution are somewhat limited. I dont see a new theoretical technique or framework coming out of this paper; it can be seen as incremental in terms of techniques.
- Presentation: The paper is not easy to read and can be better presented.
- Some parts are delegated to previous work rather than proven in the paper (e.g., "This result follows from a combination of Lemmas..."). A paper should be self-contained.
- There are inaccuracies in writing (e.g., $p$ in Algorithm 3 means two different things, etc). Please make a thorough pass before submitting a final version and make everything well-defined and correct.

---

> ### Author Rebuttal · Authors · 2026-03-31
>
> Thank you for your thoughtful review of our paper. We greatly appreciate your constructive comments and will ensure they are thoroughly addressed in the final version. Below, we respond to each of your specific questions and suggestions.
>
>
> ---
>
> > *What is the true technical novelty beyond combining known ingredients from prior work?*
>
> Our dynamic algorithm builds upon the streaming algorithm of (Alaluf et al., 2022) and the dynamic framework of (Banihashem et al., 2024) for monotone submodular maximization under matroid constraints. However, adapting and combining these methods for the non-monotone dynamic setting requires significant technical work beyond a direct combination.
> Specifically, the dynamic framework of (Banihashem et al., 2024) was designed for monotone functions and maintains only one solution set per level. In contrast, our construction requires maintaining $p = O(\epsilon^{-1})$ solutions simultaneously to handle non-monotonicity. This introduces two key challenges: (1) the analysis becomes substantially more complex, and (2) maintaining the uniform invariant — a crucial property in the original framework — is considerably more intricate.
> We believe the main contribution is showing that one can achieve, in the fully dynamic setting, the same approximation guarantee as the best streaming algorithm (up to constants), with $\text{poly}(\epsilon^{-1}, k)$ update time. This is nontrivial because streaming algorithms operate in the insertion-only model and extract a solution only at the end of the stream. In contrast, our dynamic algorithm must maintain and output a valid solution after every update, while still meeting the update time bound.
>
> Initially we considered developing a reduction that could turn any streaming algorithm into a dynamic algorithm in a black-box manner. However, we found that such a black-box reduction is not possible. The challenge lies in our level-based construction, where solutions are built consecutively across levels. To achieve the same approximation factor as the streaming algorithm in the dynamic setting, we need to maintain $p = O(\epsilon^{-1})$ solutions per level. Moreover, whenever an element in one of these solutions is deleted, or a new element is sampled and added to a solution, we must update the affected structures accordingly.
>
> Because of these dependencies, a black-box approach is insufficient. Instead, we open the streaming algorithm (i.e., adopt a white-box approach) and carefully integrate it into the dynamic framework. This allows us to bound the update time while preserving the approximation guarantee.
>
>
> > *Presentation: The paper is not easy to read and can be better presented. Are the inconsistencies only typos, or is there an actual bug you could have missed?*
>
> We thank the reviewer for this honest feedback. We acknowledge that the paper's presentation can be improved. The inconsistencies noted are indeed typos and not bugs; we have carefully verified that all proofs are correct. In the final version, we will thoroughly revise the paper to improve readability, clarify the key ideas, and polish the exposition throughout.
>
>
> > *There are inaccuracies in writing (e.g.,  in Algorithm 3 means two different things, etc). .. .
>  Could you add details on the proofs missing?*
>
> We appreciate the reviewer's careful reading. We acknowledge the issues (e.g., inconsistent use of a symbol in Algorithm 3) and will make a thorough pass to correct all inaccuracies and ensure consistent notation. Regarding the proofs, we will add the missing details. These are not errors but omissions in exposition, which we will address in the final version.
>
> > *Some parts are delegated to previous work rather than proven in the paper ...  A paper should be self-contained.*
>
> We agree with the reviewer. In the final version, we will make the paper more self-contained by including the key steps of arguments that were previously delegated to prior work, rather than simply stating that a result follows from a combination of lemmas. Essential reasoning will be presented explicitly.
>
> ---
>
>
> If you feel that our response has adequately addressed your major concerns, we would appreciate it if you could possibly adjust your score accordingly.

---

> > ### Author Rebuttal · Reviewer_hNix · 2026-04-03
> >
> > I thank the authors for their rebuttal. I stand behind my original mostly positive review.

---

> > > ### Author Response · Authors · 2026-04-08
> > >
> > > Thank you very much for your continued support and for standing by your positive review. We truly appreciate it.

---

### Official Review · Reviewer_rd84 · 2026-03-10

**Soundness:** 2
**Presentation:** 2
**Significance:** 3
**Originality:** 3
**Overall Recommendation:** 4
**Confidence:** 3

**Summary:**

This paper studies the problem of non-monotone submodular maximization with a cardinality constraint $k$ in a fully dynamic setting. It proposes a dynamic algorithm based on a randomized layered framework. The algorithm maintains $p$ parallel solutions for each guessed instance of $\mathrm{OPT}$ and incorporates a uniform invariant randomization strategy to guard against adversaries, achieving an approximation ratio of $\frac{\alpha}{1+\alpha} - \varepsilon$. When combined with the offline algorithm of Buchbinder \& Feldman (2019), the approach yields an approximation ratio of $(0.27797 - \varepsilon)$, which theoretically matches the upper bound of the best-known streaming algorithms.

**Compliance With Llm Reviewing Policy:**

Affirmed.

**Final Justification:**

In the rebuttal, the authors provided a clearer explanation regarding the novelty of their algorithm. However, the empirical results are somewhat underwhelming. Therefore, I have decided to maintain my overall positive rating of 4: Weak Accept and have raised my score for Originality from 2 to 3.

**Key Questions For Authors:**

Please refer to the weaknesses section.

**Limitations:**

Yes.

**Strengths And Weaknesses:**

**Strengths**

The paper extends the dynamic layered framework, previously developed for monotone submodular functions, to the more challenging setting of non-monotone submodular maximization. This extension is technically nontrivial and broadens the applicability of dynamic submodular optimization techniques.

By introducing a parallel maintenance mechanism with multiple solutions, the proposed algorithm achieves an approximation ratio of $\alpha/(1+\alpha)-\varepsilon$ in the fully dynamic setting. This result theoretically matches the current performance upper bound of known non-monotone streaming algorithms.

The analysis is generally well-structured, and the algorithmic framework is clearly motivated. The work contributes to bridging the gap between dynamic algorithms and streaming-style approximation guarantees for submodular maximization.

**Weaknesses**

Although the paper improves the approximation ratio for dynamic non-monotone submodular maximization to $\alpha/(1+\alpha)-\varepsilon$, the technical contribution largely builds upon existing frameworks. In particular, the proposed approach essentially combines the dynamic framework for monotone functions (Banihashem et al., 2024) with existing techniques for handling non-monotonicity (Alaluf et al., 2020). While the resulting approach is logically coherent, the level of conceptual novelty may be somewhat limited.

The paper lacks empirical validation on real-world datasets. Although the theoretical approximation guarantee is improved, the algorithm appears to incur higher update complexity compared to existing methods. Experimental results would help clarify whether this trade-off between approximation quality and computational overhead is practically worthwhile.

The proposed algorithm involves a complex hierarchical structure and relatively high query costs per update. The paper does not provide a comparison with the approach of simply rerunning a standard static algorithm after each insertion or deletion. Without such a comparison, it is difficult to assess whether the dynamic approach provides a meaningful efficiency advantage in practice.

The related work section could be further expanded. In particular, it may be beneficial to discuss deletion-robust submodular maximization algorithms (e.g., [1–3]) as well as insert-only algorithms for more general constraints (e.g., [4–7]), which are closely related to the problem setting studied in this paper.

**Minor Comments**

According to the proof of Theorem 14, the complexity result given in Lemma 12 is $O(p^2 k \log(pk) + g(pk,k))$.
Substituting $p = O(\varepsilon^{-1})$ yields
$O(\varepsilon^{-2} k \log(\varepsilon^{-1}k) + g(\varepsilon^{-1}k,k))$.
Therefore, the final bound should be
$O(\varepsilon^{-1}\log(k)(\varepsilon^{-2} k\log(\varepsilon^{-1}k) + g(\varepsilon^{-1}k,k))$.
Compared to the original statement in the paper, this expression appears to contain an additional factor of $k$. It would be helpful if the authors could clarify whether this factor is intended or whether it results from a simplification in the analysis.

[1]	Dütting P, Fusco F, Lattanzi S, et al. Deletion robust non-monotone submodular maximization over matroids[J]. Journal of Machine Learning Research, 2025, 26(66): 1-28.

[2]	Cui S, Han K, Huang H. Deletion-robust submodular maximization with knapsack constraints[C]//Proceedings of the AAAI Conference on Artificial Intelligence. 2024, 38(10): 11695-11703.

[3]	Zhang G, Tatti N, Gionis A. Coresets remembered and items forgotten: submodular maximization with deletions[C]//2022 IEEE International Conference on Data Mining (ICDM). IEEE, 2022: 676-685.

[4]	Kuhnle A. Quick streaming algorithms for maximization of monotone submodular functions in linear time[C]//International conference on artificial intelligence and statistics. PMLR, 2021: 1360-1368.

[5]	Cui S, Han K, Tang J. Linear-time algorithms for representative subset selection from data streams[C]//Proceedings of the ACM on Web Conference 2025. 2025: 4710-4721.

[6]	Chen Y, Kuhnle A. Approximation algorithms for size-constrained non-monotone submodular maximization in deterministic linear time[C]//Proceedings of the 29th ACM SIGKDD Conference on Knowledge Discovery and Data Mining. 2023: 250-261.

[7]	Cui S, Han K, Tang J, et al. Streaming algorithms for constrained submodular maximization[J]. Proceedings of the ACM on Measurement and Analysis of Computing Systems, 2022, 6(3): 1-32.

---

> ### Author Rebuttal · Authors · 2026-03-31
>
> Thank you for your thoughtful review of our paper. We greatly appreciate your constructive comments and will ensure they are thoroughly addressed in the final version. Below, we respond to each of your specific questions and suggestions.
>
> ---
>
> > Although the paper improves the approximation ratio for dynamic non-monotone submodular maximization to  , the technical contribution largely builds upon existing frameworks.
>
> We agree that our algorithm builds upon the streaming algorithm of (Alaluf et al., 2022) and the dynamic framework of (Banihashem et al., 2024) for monotone submodular maximization under matroid constraints. However, adapting and combining these methods for the non-monotone dynamic setting requires significant technical work beyond a direct combination.
> Specifically, the dynamic framework of (Banihashem et al., 2024) was designed for monotone functions and maintains only one solution set per level. In contrast, our construction requires maintaining $p = O(\epsilon^{-1})$ solutions simultaneously to handle non-monotonicity. This introduces two key challenges: (1) the analysis becomes substantially more complex, and (2) maintaining the uniform invariant — a crucial property in the original framework — is considerably more intricate.
> We believe the main contribution is showing that one can achieve, in the fully dynamic setting, the same approximation guarantee as the best streaming algorithm (up to constants), with $\text{poly}(\epsilon^{-1}, k)$ update time. This is nontrivial because streaming algorithms operate in the insertion-only model and extract a solution only at the end of the stream. In contrast, our dynamic algorithm must maintain and output a valid solution after every update, while still meeting the update time bound.
>
> Initially we considered developing a reduction that could turn any streaming algorithm into a dynamic algorithm in a black-box manner. However, we found that such a black-box reduction is not possible. The challenge lies in our level-based construction, where solutions are built consecutively across levels. To achieve the same approximation factor as the streaming algorithm in the dynamic setting, we need to maintain $p = O(\epsilon^{-1})$ solutions per level. Moreover, whenever an element in one of these solutions is deleted, or a new element is sampled and added to a solution, we must update the affected structures accordingly.
>
> Because of these dependencies, a black-box approach is insufficient. Instead, we open the streaming algorithm (i.e., adopt a white-box approach) and carefully integrate it into the dynamic framework. This allows us to bound the update time while preserving the approximation guarantee.
>
>
> ---
>
> > *The paper lacks empirical validation on real-world datasets. Although the theoretical approximation guarantee is improved, the algorithm appears to incur higher update complexity compared to existing methods. Experimental results would help clarify whether this trade-off between approximation quality and computational overhead is practically worthwhile.*
>
> We agree that empirical evaluation strengthens the paper. In response, we have conducted experiments to demonstrate the effectiveness and efficiency of our proposed algorithm, focusing on the variant that uses the offline algorithm of [1].
>
> Due to the rebuttal timeline, we performed a comparative study against the state-of-the-art algorithm by Banihashem et al. [NeurIPS 2023] on a video summarization task using real-world datasets from YouTube and the Open Video Project. We evaluated both the average number of query calls and the submodular value of the obtained solutions. Our results show that the proposed algorithm consistently achieves higher solution values compared to the baseline. A figure summarizing these results is provided at the following link: https://jumpshare.com/s/5GwjU8Mk1zHaoHV8YeTQ
>
> In the final version of the paper, we plan to expand our experiments.
>
> ---
>
> > *The related work section could be further expanded. In particular, it may be beneficial to discuss deletion-robust submodular maximization algorithms (e.g., [1–3]) as well as insert-only algorithms for more general constraints (e.g., [4–7]), which are closely related to the problem setting studied in this paper.*
>
>
> We thank the reviewer for pointing out these relevant works. We will carefully review the suggested papers and incorporate them into the related work section, along with a clearer discussion of how they compare to our approach and their limitations.
>
>
> ---
>
> > *According to the proof of Theorem 14,...*
>
> Yes, the reviewer is correct. A factor of $k$ is missing in the statement of Theorem 14. This factor is present in the proof but was accidentally omitted from the theorem statement. We will correct this typo in the final version of the paper.
>
> ---
>
>
> If you feel that our response has adequately addressed your major concerns, we would appreciate it if you could possibly adjust your score accordingly.

---

> > ### Author Rebuttal · Reviewer_rd84 · 2026-03-31
> >
> > Thank you for the rebuttal. Regarding the empirical evaluation, the results are somewhat underwhelming. The improvement in the objective function value is marginal, and it comes at the cost of a significantly higher number of oracle queries compared to prior work. Nevertheless, I still believe the theoretical contributions of this paper are solid. Therefore, I will maintain my overall positive rating and have increased my score for Originality from 2 to 3.

---

> > > ### Author Response · Authors · 2026-04-08
> > >
> > > We thank the reviewer for increasing the originality score and for the positive assessment of our theoretical contributions.  We will do more experimental work and will add the result to the paper. In the camera-ready version, we will add a discussion of the trade-off between objective improvement and oracle query cost.

---

### Official Review · Reviewer_kNdy · 2026-03-11

**Soundness:** 2
**Presentation:** 3
**Significance:** 2
**Originality:** 2
**Overall Recommendation:** 4
**Confidence:** 5

**Summary:**

To tackle the problem of fully dynamic (insertions + deletions) maximization of a non-monotone submodular function under a cardinality constraint k, this paper proposes a dynamic framework to “simulate” a strong streaming algorithm by maintaining parallel runs over guesses of the optimal solution and a randomized leveled data structure that preserves a uniform-random selection invariant to control reconstruction under deletions. Compared with previously cited dynamic guarantees ($0.125-\epsilon$ and 0.171), the main claimed improvements are approximation factors 0.262 with a stated worst-case expected update/query complexity roughly $\tilde^{O}(\epsilon^{-3} \log{k} \log(\epsilon^{-1}k) + \epsilon^{-2} k^{2} \log k)$, where $\epsilon \in (0, 1]$ is an error parameter, and 0.277 with $poly(\epsilon^{-1}, k)$ update time when using a stronger offline subroutine, aiming to match the best-known streaming factor around 0.277.

**Compliance With Llm Reviewing Policy:**

Affirmed.

**Final Justification:**

The authors have improved the presentation, and I have increased my score accordingly. Nevertheless, it is still suggested to add more experimental results for emperical validation.

**Key Questions For Authors:**

The authors are encouraged to compare with more SOTA studies on more non-monotone submodular maximization to discuss the disadvantages or insufficiencies of previous studies, especially for those size-constrained (such as [WWW25] and [KDD23]) or adaptive ones such as [ICML25].
[WWW25] Shuang Cui, Kai Han, and Jing Tang. 2025. Linear-Time Algorithms for Representative Subset Selection From Data Streams. In Proceedings of the ACM on Web Conference 2025 (WWW '25).
[KDD23] Yixin Chen and Alan Kuhnle. 2023. Approximation Algorithms for Size-Constrained Non-Monotone Submodular Maximization in Deterministic Linear Time. Proceedings of the 29th ACM SIGKDD Conference on Knowledge Discovery and Data Mining (KDD '23).
[ICML25] Chen, Y., Chen, W. &amp; Kuhnle, A.. (2025). Breaking Barriers: Combinatorial Algorithms for Non-Monotone Submodular Maximization with Sublinear Adaptivity and $1/e$ Approximation. Proceedings of the 42nd International Conference on Machine Learning, 2025.

There are some issues in the theoretical results.
Line 147: To maintain notational consistency throughout this manuscript, it would be better to replace $\Delta(e|A)$ with $f(e|A)$.
Line 382: It seems that $f(e|S_{j,i})\geq f(e|S_{j,i})$ should be replaced by $f(e|S_{j,i})\geq f(e|S_{j,T})$.
Proof of Theorem 9: It seems that the setting $\alpha\in [0,1]$ plays a crucial role in completing the proof. However, it appears that this setting was not mentioned earlier in this manuscript. Please provide further details.
Theorem 14: Compare $O(\log (k) \epsilon^{-1} (\epsilon^{-2} \log (k\epsilon^{-1}) + g(k\epsilon^(-1),k)))$ in line 428 with $O(\log (k) \epsilon^{-1} (p^2 k \log (pk) + g(pk,k)))$ in line 436, it seems that the term $O(\epsilon^{-2} \log (k\epsilon^{-1}))$ corresponds to the term $O(p^2 k \log (pk))$. However, using $p=O(\epsilon^{-1})$, we have $O(p^2 k \log (pk))=O(\epsilon^{-2} k \log (k\epsilon^{-1}))$ instead of $O(\epsilon^{-2} \log (k\epsilon^{-1}))$. Please clarify this issue.

The authors are encouraged to elaborate more details of the proposed algorithm.
The analysis explicitly uses an oblivious adversarial model for randomized invariants. The authors are encouraged to clarify how the proposed algorithm works under adaptive adversaries (which are often a concern in dynamic settings).
The algorithm has many moving parts (parallel optimal solution guesses, thresholds, p-solution levels, randomized partial reconstructions, binary search for suitability levels). The authors are encouraged to sanity-check the correctness of the algorithm and ensure no hidden dependence on the size of the given ground set n or subtle conditioning gaps with more simplifications or intuitions.

The authors are encouraged to include the experiments to show the effectiveness and efficiency of the proposed algorithm.
The authors are encouraged to include the comparative studies with the baselines listed above (see the part of [Related Work]).
The authors are encouraged to conduct sensitivity tests with different cardinality constraints $(k)$ and error parameters $(\epsilon)$.
Now that the authors design parallel runs in the proposed algorithm, the authors are encouraged to conduct efficiency tests with different parallelism degrees.

**Limitations:**

Yes. The “Impact Statement” Section is on page 9.

**Strengths And Weaknesses:**

Strengths
1. Clear progress on approximation in the dynamic non-monotone setting
2. Conceptually clean “streaming-to-dynamic” simulation
3. Thoughtful handling of deletions via randomization/uniform invariant

Weaknesses
1. More related works to be compared
2. Errors or issues in the theoretical results
3. Details of the proposed algorithm to be elaborated
4. More experiments to be enriched

---

> ### Author Rebuttal · Authors · 2026-03-31
>
> Thank you for your thoughtful review of our paper. We greatly appreciate your constructive comments and will ensure they are thoroughly addressed in the final version. Below, we respond to each of your specific questions and suggestions.
>
> ---
>
> > *The authors are encouraged to compare with more SOTA studies on more non-monotone submodular maximization to discuss the disadvantages or insufficiencies of previous studies,*
>
> We thank the reviewer for pointing out these relevant works. We will carefully review the suggested papers and incorporate them into the related work section, along with a clearer discussion of how they compare to our approach and their limitations.
>
> ---
>
> > *Q2: There are some issues in the theoretical results. Line 147, Line 382, Proof of Theorem 9, Theorem 14 and  line 428  *
>
>
> Regarding Line 147: Yes, we will change $\Delta(e \mid A)$ to $f(e \mid A)$ to resolve the inconsistency in the definition.
>
> Line 382: Thank you for catching this typo. We will correct it.
>
> Proof of Theorem 9 ("It seems that the setting $\alpha$"): We agree with the reviewer that the range for $\alpha$ was not specified earlier. While several choices of $\alpha$ are mentioned in the paragraph following Theorem 4, we will add the explicit range to the Initialize function of Algorithm 1. We will also emphasize it in the text immediately after Theorem 4.
>
> Theorem 14 (inconsistency between Lines 428 and 436): Yes, the reviewer is correct. A factor of $k$ is missing in the statement of Theorem 14. This factor is present in the proof but was accidentally omitted from the theorem statement. We will correct this typo in the final version of the paper.
>
> ---
>
> > *Q3: The authors are encouraged to elaborate more details of the proposed algorithm. The analysis explicitly uses an oblivious adversarial model for randomized invariants. The authors are encouraged to clarify how the proposed algorithm works under adaptive adversaries (which are often a concern in dynamic settings). *
>
> We think that Section 2 already outlines the main components of our dynamic algorithm. However, we agree that the presentation can be improved, and we will expand the exposition in the final version to provide a clearer overview and additional intuition behind the design.
>
> Regarding the adversarial model, our analysis assumes an oblivious adversary. As noted in Line 185 (Second Column), the use of randomness in our data structure is crucial: if the adversary could infer which elements are likely to be included in the solution, they could adapt the update sequence (insertions and deletions) to repeatedly force costly reconstructions. In our setting, the randomness helps prevent such targeted behavior, but the guarantees rely on the adversary being oblivious to the random choices.
> Handling fully adaptive adversaries—especially in dynamic submodular maximization under a cardinality constraint—remains a significant challenge. Even for monotone submodular functions, designing efficient dynamic algorithms with strong guarantees against adaptive adversaries is a major open problem.
>
>
> ---
>
> > *The authors are encouraged to include the experiments … *
>
>
> Thank you for your valuable suggestions. We agree that empirical evaluation strengthens the paper. In response, we have conducted experiments to demonstrate the effectiveness and efficiency of our proposed algorithm, focusing on the variant that uses the offline algorithm of [1].
>
> Due to the rebuttal timeline, we performed a comparative study against the state-of-the-art algorithm by Banihashem et al. [NeurIPS 2023] on a video summarization task using real-world datasets from YouTube and the Open Video Project. We evaluated both the average number of query calls and the submodular value of the obtained solutions. Our results show that the proposed algorithm consistently achieves higher solution values compared to the baseline. A figure summarizing these results is provided at the following link: https://jumpshare.com/s/5GwjU8Mk1zHaoHV8YeTQ
>
> In the final version of the paper, we plan to expand these experiments to include: (1) Sensitivity analyses with respect to different cardinality constraints and error parameters, (2) Efficiency evaluations under varying parallelism degrees, and (3) Additional comparisons with the other baselines mentioned in the Related Work section.
>
> We believe these additions will provide a comprehensive empirical assessment of our algorithm’s performance and scalability.
>
> ---
>
>
> If you feel that our response has adequately addressed your major concerns, we would appreciate it if you could possibly adjust your score accordingly.

---

> > ### Author Rebuttal · Reviewer_kNdy · 2026-04-04
> >
> > The authors have improved the presentation, and I have increased my score accordingly. Nevertheless, it is still suggested to add more experimental results for emperical validation.

---

> > > ### Author Response · Authors · 2026-04-08
> > >
> > > We would like to thank the reviewer for increasing the score. We will do more experimental work and will add the result to the paper.

---

### Official Review · Reviewer_nwQ4 · 2026-03-12

**Soundness:** 4
**Presentation:** 3
**Significance:** 4
**Originality:** 2
**Overall Recommendation:** 4
**Confidence:** 4

**Summary:**

The paper studies the non-monotone submodular maximization problem, under the regime of the cardinality constraint, in the fully dynamic setting. The authors proposed an algorithm achieving 0.277-approximation with poly(k,1/eps) update time, matching the best streaming ratio for this problem.

The key idea in my mind is to reduce a dynamic instance to a streaming one.

**Compliance With Llm Reviewing Policy:**

Affirmed.

**Final Justification:**

I appreciate the authors' response. My questions have been sufficiently answered, and I remain supportive of the paper’s acceptance.

**Key Questions For Authors:**

1. Why do you state it as an algorithm, not a reduction? Are these two ratios (for streamming and dynamic problems) always equal in your "algorithm"?
2. It would be great if you have any example how updates propogate?

**Limitations:**

yes

**Strengths And Weaknesses:**

Strengths:
1. This paper close the gap between the ratios of dynamic and stream setting for the problem.

Weaknesses:
1. The first sentence is more-or-less redudant in the abstract of ICML submission.
2. The "algorithm" heavily rely on the existing mehtods in streamming setting.

---

> ### Author Rebuttal · Authors · 2026-03-31
>
> We thank the reviewer for the careful reading of our paper and for the constructive feedback and thoughtful questions.
>
> ---
>
> > *W1: The first sentence is more-or-less redudant in the abstract of ICML submission.”
>
> We thank the reviewer for this observation. We agree that the first sentence ("Non-monotone submodular maximization is a fundamental problem in machine learning and combinatorial optimization...") may be too general for an ICML abstract. In the final version, we will remove or significantly condense this sentence to focus directly on the problem setting and our contributions. We will replace it with a more concise lead-in that immediately establishes the fully dynamic, non-monotone, cardinality-constrained setting and our improved guarantees.
>
> ---
>
> > *W2: The "algorithm" heavily rely on the existing mehtods in streamming setting.*
>
> Thank you for this observation. We agree that our algorithm builds upon the streaming algorithm of (Alaluf et al., 2022) and the dynamic framework of (Banihashem et al., 2024) for monotone submodular maximization under matroid constraints. However, adapting and combining these methods for the non-monotone dynamic setting requires significant technical work beyond a direct combination.
> Specifically, the dynamic framework of (Banihashem et al., 2024) was designed for monotone functions and maintains only one solution set per level. In contrast, our construction requires maintaining $p = O(\epsilon^{-1})$ solutions simultaneously to handle non-monotonicity. This introduces two key challenges: (1) the analysis becomes substantially more complex, and (2) maintaining the uniform invariant — a crucial property in the original framework — is considerably more intricate.
>
> We believe the main contribution is showing that one can achieve, in the fully dynamic setting, the same approximation guarantee as the best streaming algorithm (up to constants), with $\text{poly}(\epsilon^{-1}, k)$ update time. This is nontrivial because streaming algorithms operate in the insertion-only model and extract a solution only at the end of the stream. In contrast, our dynamic algorithm must maintain and output a valid solution after every update (both insertions and deletions), while still meeting the update time bound.
>
>
> ---
>
> > *Q1: Why do you state it as an algorithm, not a reduction? Are these two ratios (for streamming and dynamic problems) always equal in your "algorithm"?*
>
> Thank you for this thoughtful question. Initially, we also considered developing a reduction that could turn any streaming algorithm into a dynamic algorithm in a black-box manner. However, we found that such a black-box reduction is not possible.
> The challenge lies in our level-based construction, where solutions are built consecutively across levels. To achieve the same approximation factor as the streaming algorithm in the dynamic setting, we need to maintain $p = O(\epsilon^{-1})$ solutions per level. Moreover, whenever an element in one of these solutions is deleted, or a new element is sampled and added to a solution, we must update the affected structures accordingly.
>
> Because of these dependencies, a black-box approach is insufficient. Instead, we open the streaming algorithm (i.e., adopt a white-box approach) and carefully integrate it into the dynamic framework. This allows us to bound the update time while preserving the approximation guarantee. Thus, what we present is a fully specified algorithm rather than a reduction.
> Regarding the ratios: In our construction, the approximation factor of the dynamic algorithm matches that of the underlying streaming algorithm up to constant factors. The equality is not automatic — it is achieved through the careful white-box integration described above.
>
>
> ---
>
> > *Q2: It would be great if you have any example how updates propogate?*
>
> If we understand the question correctly, our solution is as follows.
> Consider an instance with $k=2$, $p=2$, and three levels: $e_1=a$ (to $S_{1,1}$), $e_2=b$ (to $S_{1,2}$), $e_3=c$ (to $S_{2,3}$). Upon deletion of $b$, the algorithm iterates over levels. At $i=1$, $e_1 \neq b$, so $b$ is removed from $R_1$ (if present). At $i=2$, $e_2 = b$, so $b$ is removed from $R_2$ and reconstruction is triggered from level $2$ onward.
>
> The reconstruction uses the updated $R_2 = R_2^- \setminus {b}$. A random permutation of $R_2$ is generated; the first suitable element becomes the new $e_2$. For instance, if $R_2^- = {b,d,e}$, then $R_2 = {d,e}$. If $d$ is suitable for $S_{1,1}={a}$, it becomes $e_2$. Reconstruction then proceeds to subsequent levels. Only levels $\geq 2$ are rebuilt; level $1$ remains unchanged. Since $T \leq pk = O(\epsilon^{-1}k)$ and each suitability check takes $O(p)$ time, the update time is $\operatorname{poly}(\epsilon^{-1}, k)$.
>
>
>
>
> ---
>
>
> If you feel that our response has adequately addressed your major concerns, we would appreciate it if you could possibly adjust your score accordingly.

---

> > ### Author Rebuttal · Reviewer_nwQ4 · 2026-04-06
> >
> > I appreciate the authors' response. My questions have been sufficiently answered, and I remain supportive of the paper’s acceptance.

---

> > > ### Author Response · Authors · 2026-04-08
> > >
> > > We once again thank the reviewer for reading our paper and for providing insightful comments.

---

### Decision · Program_Chairs · 2026-04-30

**Decision:**

Accept (regular)

**Comment:**

There is a nice line of work studying non-monotone submodular maximization subject to a cardinality constraint in a fully dynamic setting:
- $(0.125 - \varepsilon)$-approximation algorithm [Banihashem et al., NeurIPS 2020]
- $(0.171 - \varepsilon)$-approximation algorithm [Banihashem et al., NeurIPS 2025]

This work presents an $(0.262 - \varepsilon)$-approximation for the same problem with worst-case expected update time $\tilde{O}(\varepsilon^{-3} + \varepsilon^{-2} k^2)$.

**Decision.** All reviewers recommended "Weak Accept," and I recommend the same.